# Occupational Characteristics in the Outbreak of the COVID-19 Delta Variant in Nanjing, China: Rethinking the Occupational Health and Safety Vulnerability of Essential Workers

**DOI:** 10.3390/ijerph182010734

**Published:** 2021-10-13

**Authors:** Yujun Liu, Bowen Yang, Linping Liu, Maitixirepu Jilili, Anuo Yang

**Affiliations:** 1Department of Social Work and Social Policy, School of Social and Behavioral Sciences, Nanjing University, Nanjing 210023, China; yujunliu@nju.edu.cn; 2Department of Sociology, School of Social and Behavioral Sciences, Nanjing University, Nanjing 210023, China; yangbowen0717@163.com (B.Y.); dg1907012@smail.nju.edu.cn (M.J.); dg1907020@smail.nju.edu.cn (A.Y.)

**Keywords:** COVID-19 disease, Delta variant strain, Nanjing Lukou International Airport, airport cleaner, essential worker, infection risk, occupational characteristics, occupational health and safety vulnerability, epidemic prevention and control

## Abstract

The risk of contracting COVID-19 varies by occupation. Clarifying the occupational disparity in the infection risk is crucial to the prevention and control of the epidemic in the workplace. In late July, some new cases of COVID-19 were confirmed among cleaners working in Lukou International Airport in Nanjing, China. The infected cases rapidly increased and spread to many domestic cities in the following days. The present study traces the brief reports of epidemiological investigations among the confirmed cases released by the Nanjing government from 20 July to 2 August, and offers a descriptive analysis on the occupational distribution of these cases. Cleaners and other staff working in the airport were found to make up more than 40% of all cases. The overwhelming majority of the cleaner cases were confirmed in the first 7 days. The present study statistically ascertains that the airport cleaners were the initial sufferers and transmitters in this outbreak. They experienced occupational health and safety vulnerability on both individual and contextual levels, including workplace hazards, workplace safety policies, and lack of awareness and empowerment. Effective protection for essential workers and the strict surveillance of occupational health in the workplace is urgently needed.

## 1. Introduction

### 1.1. COVID-19 Outbreak in Nanjing

*Nanjing Fa Bu*, the new media platform of the Municipal Party Committee and the Municipal government of Nanjing, which is the provincial capital of Jiangsu, China, and has a population of more than 9.3 million, notified that nine samples of the regular nucleic acid testing among the staff working in Lukou International Airport tested positive for SARS-CoV-2 on 20 July. On the 21st, the Nanjing government announced that there were seven new local confirmed cases and two asymptomatic cases [1]. These cases were all female cleaners working in Lukou International Airport, aged from 38 to 51. On the same day, the government delimited four medium-risk areas and one control area and decided to carry out the first nucleic acid testing among all permanent-resident populations and visitors in Nanjing. The number of new local confirmed cases increased rapidly in the following days and peaked on the 27th (47 cases). As of 24:00 on 2 August, the total number of local confirmed cases in Nanjing was 220, including 82 light cases, 132 regular cases and 6 severe cases. More seriously, confirmed cases related to this outbreak were identified in many cities such as Yangzhou, Huaian and Zhangjiajie (in Hunan Province).

On 30 July, the Nanjing government held the tenth press conference on this COVID-19 outbreak. In the conference, an official of Nanjing Municipal Center for Disease Control and Prevention reported the finding of virus genes in 52 related cases in this outbreak [2]. The virus genome sequences were highly homologous, indicating the same transmission chain. The early cases (airport cleaners) have been found to share the same RNA sequence as the Delta variant strain, which was consistent with the genetic sequence of an imported case on Flight CA910 from Russia on 10 July. The official suggested that several airport cleaners may have been infected due to the nonstandard washing and removing of protective equipment after the cabin cleaning on Flight CA910. Then, the virus was transmitted among the employees in the cleaning company to which the airport cleaners belonged. Furthermore, because those cleaners were responsible for the garbage clearance of international and domestic flights simultaneously, other staff working in the airport might have been infected, by contact with those cleaners or the contaminated environment.

An anonymous cleaner working in Lukou International Airport, who was a confirmed case, provided more details in an interview [3]. The cleaners usually sprayed the disinfector and cleaned the cabin immediately after the passengers left, without closing the door to allow the disinfector to react in the confined space for a sufficient amount of time. The cleaners took part in the cleaning of international and domestic flights concurrently, and shared the cleaning tools such as brooms, duster cloths, dust collectors, and the rest room. Furthermore, they were required to carry the cabin garbage under the airbridge, which may have led to the secondary pollution of the garbage. 

Information from multiple sources revealed that the nonstandard methods of disinfection and the mismanagement of Lukou International Airport were the key factors leading to this COVID-19 outbreak in Nanjing. Some news agencies pointed out that the airport outsourced its cleaning business to external cleaning companies in 2019 [4,5]. Although outsourcing is a common practice in airport management, the work of the contractor needs to be surveilled by the airport. The prevention of the epidemic in cleaning work is the most important component of the whole epidemic-prevention system in the airport. Given that the early cases in this outbreak were among the cleaners and other staff working in the airport, and the Delta variant is highly transmissible, the occupational risk in the transmission of SARS-CoV-2 virus is alarming. 

### 1.2. Occupational Risk in Contracting COVID-19

For some people, the workplace is not only a place to earn payment and achieve personal value, but also a place which is inundated with risks that threaten their physical and mental health. Previous empirical studies have revealed that the work organization and job characteristics, such as work schedules, psychosocial job stressors and the organizational climate, are significantly related to the workers’ health conditions [6,7]. However, the distribution of occupational health risks is uneven. Workers in some occupations encounter more health hazards than those in other occupations, and confront specific risks in special periods or contexts. For instance, a Belgian census-linked mortality study reported that respiratory and cardiovascular mortality is considerably higher for male and female cleaners than for non-manual workers [8]. 

Existing studies found that an individuals’ exposure to infectious diseases is closely associated with their occupation [9,10,11]. The occupational risk in COVID-19 infection has been demonstrated in several studies. At the very beginning of the outbreak, people who worked in the Huanan Seafood Wholesale Market in Wuhan were a high-risk group [12]. Then, the healthcare workers (HCWs) such as the doctors, nurses and paramedical staff that were working at the front line of the global pandemic and confronting substantial risk in workplace, obtained the most academic attention. A systematic review showed that a total of 152,888 infections and 1413 deaths were reported among HCWs worldwide in 2020 [13]. A study conducted in Wuhan reported that the case infection rate of HCWs (2.10%) was dramatically higher than that of non-HCWs (0.43%) [14]. 

The risk of COVID-19 infection in other occupations seemed to receive less attention. A handful of studies summarized the infection status across different occupations. For instance, Lan and colleagues observed work-related COVID-19 transmission in six Asian countries/areas for 40 days after its first locally transmitted case [15]. They identified five occupation groups with the most cases: healthcare workers, drivers and transport workers, services and sales workers, cleaning and domestic workers and public safety workers in proper order. Feng and colleagues described the occupational characteristics of the confirmed cases (such as occupations related to the cold-chain) in sporadic COVID-19 outbreaks from June to December 2020 in China [16]. They suggested that the occupation and work information were crucial in tracking down “patient zero” and reducing the transmission rate. 

COVID-19 is the first new occupational disease to be described this decade [17]. The outbreak in Nanjing, the source of which was the infected cleaner who cleaned the cabin of Flight CA910 that subsequently reported an imported case, brings our attention back to the occupational risk and vulnerability during viral transmission. As an important pivot of the world-class airport group in the Yangtze River Delta, Nanjing Lukou International Airport has air routes to 78 domestic cities and 35 international destinations, and the passenger throughput for the year was 25.8 million in 2017 [18]. In a place with such a huge flow of people, why did the COVID-19 Delta variant transmit first among airport cleaners, rather than the passengers or flight attendants? Do the individual characteristics or the surveillance system in the airport make cleaners more vulnerable to COVID-19 infection? How many cleaners were infected and what was the proportion? These questions need to be clarified to implement effective protection in the workplace, prevent the epidemic, and control rapid transmission of the Delta variant. 

In the present study, a conceptual model developed by Smith and colleagues was employed to address the occupational disparity in the risk of being infected in the Nanjing outbreak [19]. This model defined four dimensions of occupational health and safety (OHS) vulnerability, including: (1) exposure to workplace hazards; (2) workplace safety policies and procedures; (3) worker awareness of health and safety-related rights and responsibilities; and (4) worker empowerment to act to protect themselves and colleagues. The core idea of this model was that vulnerability resulted from exposure to on-the-job hazards, along with inadequate access to resources (policies and procedures, awareness or empowerment) to mitigate the effects of these risks. In other words, workers’ vulnerability was not only determined by their personal traits but was also related to their working environment. The OHS vulnerability model was utilized in research on multiple working populations, such as recent immigrants and refugees, brick-factory workers and nonstandard workers [20,21,22]. Considering the similarities of the working conditions between these workers and the airport cleaners, we adopted this model in our present study.

Taken together, the objective of this study was to describe the occupational characteristics among confirmed cases of the COVID-19 outbreak in Nanjing and investigate the potential factors leading to OHS vulnerability on both an individual and contextual level. Policies and the practical implications of maintaining and promoting the occupational health and safety of essential workers in the midst of the pandemic are also discussed. 

## 2. Methods

Since the first batch of new local cases in Nanjing was identified on 20 July, *Nanjing Fa Bu*, the official media platform, has released the information of epidemiological investigations into new confirmed cases on a daily basis, to help citizens check the history of contact. We followed the information on this platform from 21 July to 3 August (confirmed cases from 20 July to 2 August), because 14 days is widely believed as the incubation period suggested by health authorities for active monitoring [23]. The data we recorded and summarized included: each confirmed case’s gender, age, occupation, the date, and the type of symptom when diagnosed. It should be noted that we did not collect the individuals’ characteristics directly from the confirmed cases; the anonymous information was extracted from the epidemiological investigation reports issued by the Nanjing government. The number of cases that we included in our final analysis was 220. 

Confirmed cases in the present study fell into 5 occupational groups, including: airport cleaners, other staff working in the airport (such as airline ground staff, auxiliary police officers, drivers and restaurant staff), peasants/retirees/the unemployed, other occupations, and children/adolescents. Given that the occupations of those cases who did not work in the airport were extremely varied, we categorized them in the “other occupation” group. Preschool children and students (aged from 8 months to 18 years) were classified as “children and adolescents” in our analysis. 

Descriptive analyses were performed by R (a free language and environment for statistical computing and graphics). We used the variance analysis to compare the mean ages of five occupational groups. As the cases in one category were less than five, the Fisher’s precision probability test was adopted to compare the gender proportion across five occupations. Chi-squared tests were included to examine the differences of notification dates and symptoms among occupational groups. Multiple comparisons using the Bonferroni approach were utilized to uncover if there were significant differences on these characteristics between any two occupational groups. The figure was plotted by Excel (Microsoft, Redmond, WA, USA).

## 3. Results

The basic characteristics of the cases are summarized in Table 1. The mean age was 43.25 (SD = 16.96) and females made up nearly 60% of all cases. The 69 airport cleaners accounted for more than 30% of all cases. The proportion of other staff working in the airport was 13.18%. Almost 11% of cases were children and adolescents. As a result of the Nanjing government implementing the second and the third nucleic acid tests on 25 July and 28, respectively, confirmed cases reported in the second 7 days (27 July–2 August) were slightly higher than those reported in the first 7 days (20 July–26 July). More than 60% of the cases had light symptoms when they were diagnosed.

Table 2 displays the basic characteristics of confirmed cases grouped by five occupations. There were statistically significant differences in age, gender and date of notification across the different occupational groups, indicating a necessity to further conduct the comparisons between any two occupational groups. However, symptoms when being diagnosed were not significantly related to occupation.

Multiple comparisons were performed using the Bonferroni approach. We paid particular attention to the airport cleaners and other staff working in the airport. The same subscripted letter “a” in the first row in Table 2 meant that the difference between the mean age of the airport cleaners (44.68) and of other staff working in the airport (40.62) or other occupations (43.74) was not statistically significant. The proportion of female airport cleaners was 91.3%, which was significantly higher than that of other staff working in the airport (14.29%) or in other occupations (44%). Accordingly, the overwhelming majority of other staff working in the airport were male. Moreover, compared to the other four occupational groups, the vast majority of the airport cleaner cases were confirmed or notified in the first 7 days of transmission (76.81%). There were not significant differences in the date of notification among the other four occupations.

The trends in new local cases confirmed daily from 20 July to 2 August are displayed in Figure 1. The new cases peaked on 27 July and declined over the next few days. The number of airport cleaner cases, other staff working in the airport and other confirmed cases are marked in Figure 1. In the first 7 days of transmission, airport cleaner cases accounted for a large proportion of daily new confirmed cases. Although this proportion fluctuated, the overall trend declined after 14 days. The proportion of other airport staff in daily new confirmed cases peaked on 24 July, then declined with small fluctuations.

## 4. Discussion

In the present study, we collected the personal information of the confirmed daily cases from 20 July to 2 August in Nanjing, including gender, age, occupation, date and symptoms when diagnosed. We first provided a description of the confirmed cases’ occupational characteristics, and then preliminarily examined the differences among these occupational groups. We found that nearly one-third of confirmed cases were airport cleaners, and that the overwhelming majority of cleaner cases were confirmed in the first 7 days of this outbreak. Other staff working in the airport accounted for around 13% of all cases. Later transmission occurred mainly in cleaners’ households and communities near the airport, without a significant correlation with occupation. We statistically ascertained that the airport cleaners were the initial sufferers and early transmitters in this outbreak.

We argue that the disproportionate infection among airport cleaners in this outbreak reflects the occupational health and safety vulnerability of this occupation. As mentioned in previous paragraphs, the cleaners were responsible for the cleaning and disinfection of domestic and international flights concurrently, and were required to carry the cabin garbage to the lounge bridge by themselves. Considering that the daily new confirmed cases in foreign countries were far more than that in mainland China, the duties related to the international flights placed the cleaners at a high risk of infection. Furthermore, although an anonymous airport cleaner disclosed that they were asked to wear protective suits when cleaning the international flights and were trained to wear the personal protective equipment in the correct order [3], this outbreak exposed the defects of management and surveillance for the cleaning and daily functioning of the airport. Some media have revealed that the cleaning business in Lukou Airport was outsourced to an external cleaning company. In general, the cleaning company should be responsible for the employment and training of their cleaners, whereas the airport is not involved in the company’s management system. It is reported that the airport cleaners were asked to simultaneously clean the domestic and international flights because the cleaning company wanted to save costs, and the airport failed to perform its surveillance duty [5]. 

This outbreak reflects the drawbacks of labor outsourcing, which emerged at the end of the 20th century in China. Limited by the gross payroll system, many large state-owned enterprises, such as Lukou Airport, outsource their ancillary services, such as the cleaning service, to external specialized agencies. In this employment mode, the airport is the contractee, and the cleaning company is the contractor; there is not a legal labor relation between the airport and the cleaners. Labor outsourcing is considered as a form of flexible employment and is expected to save operating costs; however, it could also harm the employees’ rights, and put them at risk. In the case of Lukou Airport, the cleaners were the main victims of mismanagement.

In addition to the vulnerabilities of workplace hazards and workplace safety policies, the airport cleaners in the Nanjing outbreak experienced another vulnerability: lack of awareness towards health and safety in the workplace. The anonymous airport cleaner mentioned above recalled that a co-worker coughed on July 13 and infected several colleagues. Nevertheless, this phenomenon did not attract the attention of the cleaners and administrative staff in the cleaning company, which may foreshadow the later transmission. We contend that inadequate awareness arises from individual characteristics in conjunction with contextual factors. On one hand, the cleaning service is a labor-intensive industry, and does not require a good educational background. The epidemiological investigation also showed that most of the cleaners were middle-aged female villagers living near the airport, whose health literacy may have limited their awareness to the infection risk. On the other hand, to some extent this can be regarded as the negative result of chronically neglecting the occupational health and safety among essential workers.

Essential workers are those who work in health care, cleaning services, delivery services, retail establishments, agriculture and other essential industries. Telework, widely adopted by teachers and programmers in the pandemic, is not applicable among essential workers. In other words, society is not able to function normally without essential workers. The indispensable nature of essential work means that workers are subjected to elevated health hazards, as well as concerns about transmitting the virus to their family and community members [24]. As a matter of fact, there appeared a cluster of cases in the Sixth People’s Hospital in Zhengzhou, the capital city of Henan Province since 31 July, of which the earliest cases included two cleaners in this hospital. On 2 and 5 August, Shanghai Pudong Airport and Haikou Meilan Airport separately reported one service staff and one stevedore who tested positive for nucleic acid. On 4 September, a female cleaning staff member working in a quarantine hotel in Guangzhou, the capital city of Guangdong Province, tested positive for nucleic acid. The essential workers in airports, hospitals and hotels, who confront an impaired health risk in the workplace in ordinary times, have become the high-risk group in the COVID-19 epidemic.

Although the Labor Law of the People’s Republic of China has a section on labor health and safety, essential workers, especially those who migrate from rural areas to cities and work in 3D (dirty, difficult and dangerous) industries, suffer from huge health hazards [25]. These essential workers usually work in a detrimental environment without effective protective equipment or work overtime frequently. Additionally, most of them do not have social and medical security. Moreover, this population experiences power-related vulnerability; they do not have a large voice, leading to their health conditions becoming marginalized and invisible. For instance, the cleaners and care workers in hospitals are exposed to higher infection risks because they are in direct contact with patients and medical waste, and their awareness and knowledge is not as sufficient as medical professionals. They do not receive the attention they deserve.

After a series of strict prevention and control measures, there have been no new cases in Nanjing since 13 August, and the whole city has been designated as a low-risk area since 19 August. Lukou Airport also resumed domestic flights on 26 August after the completion of terminal disinfection in all areas. This Delta-variant COVID-19 outbreak raises new challenges to epidemic prevention and control in China and the global community. In the meantime, it urges the authorities and employers to seriously rethink the occupational health and safety vulnerability of essential workers. As Baker suggested, surveillance of occupational health might be a powerful tool for COVID-19 prevention [26]. On one hand, in workplaces such as airports and hospitals, in which the environment is complicated and the number of people is huge, the health and safety of essential workers should receive equal, if not more attention, than other workers. Periodic screening programs, sufficient personal protective equipment and occupational health education to raise awareness of the risks should be provided [27]. On the other hand, although the essential workers in these workplaces may be directly managed by outsourcing companies, the contractee (the airport or hospital) should routinely perform surveillance to ensure the well-being of their workers.

Although the present study illustrates the occupational health and safety vulnerabilities in the COVID-19 outbreak in Nanjing for the first time, it has certain limitations. Most of all, as mentioned in the methods section, we collected the individual characteristics of confirmed cases in this outbreak from the official reports of epidemiological investigations carried out by the government, instead of interviewing them face-to-face. Therefore, the information only contained their age, gender, occupation, date of notification and symptoms when diagnosed. More detailed socio-demographic characteristics and clinical features (such as a change in symptoms, the course of illness and follow-up rehabilitation) are unknown to us. Furthermore, the sample in the present study only covered the sectional confirmed cases in the Nanjing outbreak. We are unable to include further classification of occupations and cannot compare the morbidity and mortality rates across more occupational groups in the analysis. In other words, the small-sized, information-limited and cross-sectional data prevented us from performing causal inference to inquire the potential causal relations between occupation and the infection risk of COVID-19. More scientific and longitudinal research designs are needed to address this gap in the future.

## 5. Conclusions

In conclusion, the present study describes the occupational characteristics among confirmed cases of the Delta-variant COVID-19 outbreak in late July in Nanjing. Using the data extracted from the brief reports of epidemiological investigation released by the Nanjing government, we statistically confirmed that the airport cleaners were the initial victims and early transmitters of this outbreak. We analyzed the individual and contextual factors that led to the vulnerability of airport cleaners in terms of occupational health and safety, and demonstrated that labor outsourcing and the consistently unheeded occupational health and safety among essential workers were the source of this transmission. We urge the government and employers to rethink this issue, not only to prevent and control the COVID-19 epidemic, but also to establish a healthy workplace and improve the workers’ well-being.

## Figures and Tables

**Figure 1 ijerph-18-10734-f001:**
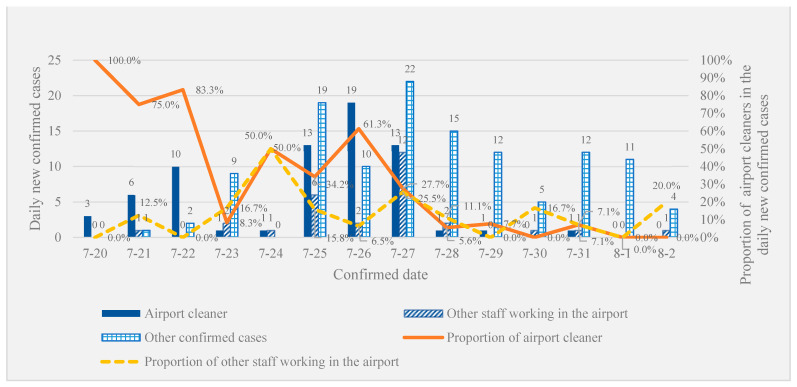
The trends of new local cases confirmed daily from 20 July to 2 August.

**Table 1 ijerph-18-10734-t001:** Basic Characteristics of Confirmed Cases of COVID-19 in Nanjing from July 20 to August 2 in 2021 (N = 220).

Variable	Mean (SD) or N (%)
Age	43.25 (16.96)
Gender	
Female	131 (59.82)
Male	88 (40.18)
Occupation	
Airport cleaner	69 (31.36)
Other staff working in the airport	29 (13.18)
Peasant, retiree and unemployed person	48 (21.82)
Other occupations	50 (22.73)
Children and adolescents	24 (10.91)
Date of notification	
The first 7 days	106 (48.18)
The second 7 days	114 (51.82)
Symptoms	
Light	137 (62.27)
Regular	83 (37.73)

**Table 2 ijerph-18-10734-t002:** Basic Characteristics of Confirmed Cases Categorized by Occupation in Nanjing.

	Airport Cleaner	Other Staff Working in the Airport	Peasant, Retiree and Unemployed Person	Other Occupations	Children and Adolescents	*p* Value
Age	44.68 (5.35) _a_	40.62 (9.54) _a_	59.19 (15.20) _b_	43.74 (11.79) _a_	9.45 (6.01) _c_	0.0000
Gender						
Female	63 (91.30) _a_	4 (14.29) _b_	31 (64.58) _c_	22 (44.00) _b,c_	11 (45.83) _b,c_	0.0005
Male	6 (8.70) _a_	24 (85.71) _b_	17 (35.42) _c_	28 (56.00) _b,c_	13 (54.17) _b,c_	
Date of notification						
The first 7 days	53 (76.81) _a_	12 (41.38) _b_	16 (33.33) _b_	19 (38.00) _b_	6 (25.00) _b_	0.0000
The second 7 days	16 (23.19) _a_	17 (58.62) _b_	32 (66.67) _b_	31 (62.00) _b_	18 (75.00) _b_	
Symptoms						
Light	45 (65.22) _a_	18 (62.07) _a_	27 (56.25) _a_	29 (58.00) _a_	18 (75.00) _a_	0.5512
Regular	24 (34.78) _a_	11 (37.93) _a_	21 (43.75) _a_	21 (42.00) _a_	6 (25.00) _a_	

Note: Subscripted lowercase letters (a, b, c) represent the subset of categories among occupations. The same letters in different categories suggest that there are not statistically significant differences, and different letters in different categories indicate that there are significant differences (*p* < 0.05).

## Data Availability

The datasets analyzed for this study can be found in *Nanjing Fa Bu* (https://mp.weixin.qq.com/s/i-fqS2-5AWV--oPE_Zl-Uw, accessed on 6 August 2021).

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
