# Peer review of "Occupational Characteristics in the Outbreak of the COVID-19 Delta Variant in Nanjing, China: Rethinking the Occupational Health and Safety Vulnerability of Essential Workers"

_ijerph, 2021, doi:10.3390/ijerph182010734_

Round 1
Reviewer 1 Report
The present manuscript is a brief report about the vulnerability to be infected by COVID-19 depending on the occupation, particularly in the province of Naijing (China).
Despite the fact that is a brief report, more information are needed to methods section. In methods section you must explain about the type of analysis and the tests used, it is explain in paragraphs 134-146, but it must change to methods section. Also, it's important that the authors write the objectives of the study at the end of the introduction.
Please can you explain why in Table 1 did you provide frequency about "airport cleaner", "children and adolescents". I understand the focus of the present study is on occupations. I don't understand this approach.
Table 2. Can you explain the significance of the last colum (p value)? This p value to which association did you refer to? I mean, a p value is obtain through all the associations (age-airport cleaner, age-other staff...).
If the tittle is "occupational vulnerability" but you only focus on airport cleaner and other staff of the airport. You must indicate that you are focused on airport professionals. It's important that the authors delimitated the objectives to understand the main idea. Also, please in methods section explain the study context in more detail.
Conclusions. A conclusion is that the cases infected with Delta variant. It should be included in the tittle, objectives or anywhere, but previous conclusions section.
I don't understand why "informed consent statement" is not applicable. You can obtain the personal data of people without their consent? How can did you do this?
References. Please provide DOI or link of the articles. Review if the references are corrected by format.
Author Response
Dear Reviewer,
Thanks very much for all of your valuable advices. Our responses and revisions please see in the response letter and new manuscript.
Kind regards,
The Authors

Reviewer 2 Report
I read the article in question with interest. I believe that its content can be important to readers and can focus on important risks. However, I believe that improvements can be made, so I indicate some suggestions that are useful in my opinion:
- the problem of outsourcing is well explained but I believe other useful elements can be added that show a) the reasons why these workers were more at risk of infection and b) the possibilities of preventing contagion.
- the limits of the study can be highlighted more clearly, for example with regard to the number of samples and the way in which the results are read
- the bibliography is poor and needs more insertion of appropriate international articles. I can suggest a few but the authors may endeavor to do more research on this point. ex. doi: 10.3390 / ijerph18052484.
doi: 10.3390 / healthcare9010017.
doi: 10.4103 / ajm.ajm_171_20. eCollection 2020 Oct-Dec.
Author Response

(The authors gave the same response as above.)

Reviewer 3 Report
Thank you for an interesting paper. While I see the overall merit of the paper, I feel that it is not sufficiently linked to the topic of occupational health. Here are a few suggestions:
- the literature review should be expanded. The authors talk about previous studies, but do not provide sufficient evidence of previous findings
- The topic of occupational vulnerability is interesting, but is hardly mentioned in the paper. It would also be useful to provide a proper definition of the term and - if applicable - include previous studies dealing with the concept.
- Method: Why was this particular time frame chosen for examination? Please also provide more details on the data collection - so far, it is very vague.
- The paper itself should be embedded in the broader context of occupational health - so far, hardly any relevant concepts and/or models are discussed. (even though the title suggests otherwise)
- The title talks about "rethinking" occupational health. For me, this implies that some implications will be derived. Please consider this aspect in the revision.
I hope you find the feedback helpful!
Author Response

(The authors gave the same response as above.)

Round 2
Reviewer 1 Report
Dear authos, thank you very much for your clarifications and improvements of the document. I consider the present manuscript to be accepted for publication.
Reviewer 2 Report
The authors have worked hard and sufficiently improved the text, in my opinion it is now publishable